# Correlation of Serum N-Acetylneuraminic Acid with the Risk of Moyamoya Disease

**DOI:** 10.3390/brainsci13060913

**Published:** 2023-06-05

**Authors:** Chenglong Liu, Peicong Ge, Chaofan Zeng, Xiaofan Yu, Yuanren Zhai, Wei Liu, Qiheng He, Junsheng Li, Xingju Liu, Jia Wang, Xun Ye, Qian Zhang, Rong Wang, Yan Zhang, Jizong Zhao, Dong Zhang

**Affiliations:** 1Department of Neurosurgery, Beijing Tiantan Hospital, Capital Medical University, 119 South Fourth Ring West Road, Fengtai District, Beijing 100070, China; 18976866841@163.com (C.L.); gepeicong@163.com (P.G.); zchf723@163.com (C.Z.); xfyumail@gmail.com (X.Y.); 18610638233@163.com (Y.Z.); liuwei980227@163.com (W.L.); heqiheng96@mail.ccmu.edu.cn (Q.H.); lijunsheng723@163.com (J.L.); liuxingju006@163.com (X.L.); wangjia1118@hotmail.com (J.W.); yexun@bjtth.org (X.Y.); zhangqianchina@yahoo.com (Q.Z.); ronger090614@126.com (R.W.); yanzhang135@163.com (Y.Z.); 2China National Clinical Research Center for Neurological Diseases, Beijing 100070, China; 3Center of Stroke, Beijing Institute for Brain Disorders, Beijing 100070, China; 4Beijing Key Laboratory of Translational Medicine for Cerebrovascular Disease, Beijing 100070, China; 5Beijing Translational Engineering Center for 3D Printer in Clinical Neuroscience, Beijing 100070, China; 6Department of Neurosurgery, Beijing Hospital, Beijing 100730, China

**Keywords:** moyamoya disease, N-acetylneuraminic acid, metabolites, biomarkers, cerebrovascular disorders

## Abstract

N-acetylneuraminic acid (Neu5Ac) is a functional metabolite and has been demonstrated to be a risk factor for cardiovascular diseases. It is not clear whether Neu5Ac is associated with a higher risk of cerebrovascular disorders, especially moyamoya disease (MMD). We sought to elucidate the association between serum Neu5Ac levels and MMD in a case–control study and to create a clinical risk model. In our study, we included 360 MMD patients and 89 matched healthy controls (HCs). We collected the participants’ clinical characteristics, laboratory results, and serum Neu5Ac levels. Increased level of serum Neu5Ac was observed in the MMD patients (*p* = 0.001). After adjusting for traditional confounders, the risk of MMD (odds ratio [OR]: 1.395; 95% confidence interval [CI]: 1.141–1.706) increased with each increment in Neu5Ac level (per μmol/L). The area under the curve (AUC) values of the receiver operating characteristic (ROC) curves of the basic model plus Neu5Ac binary outcomes, Neu5Ac quartiles, and continuous Neu5Ac are 0.869, 0.863, and 0.873, respectively. Furthermore, including Neu5Ac in the model offers a substantial improvement in the risk reclassification and discrimination of MMD and its subtypes. A higher level of Neu5Ac was found to be associated with an increased risk of MMD and its clinical subtypes.

## 1. Introduction

Moyamoya disease (MMD) is an abnormal cerebrovascular disease identified by progressive stenosis, occlusion of the large arteries around the Willis circle, and compensatory development of certain blood vessels in the basal brain. In the course of cerebral angiography, the consequent irregular vascular network in the brain is unclear and smoky, hence the name [1,2]. Although MMD is a cerebrovascular disease with a low incidence, it is the leading cause of stroke in children and adolescents in Asia, including Korea, Japan, and China [3,4]. Since traditional risk factors cannot fully explain all MMD cases, increasing attention has been paid to identifying novel, highly modifiable risk factors to predict the risk of MMD, reduce the severity of MMD, and improve its prognosis.

In recent years, variations in the composition of gut microbiota in the body and its metabolic changes have been recognized as essential factors in the occurrence and development of cardiovascular disease [5,6,7]. There is increasing evidence that both genetic and environmental factors, as well as the interactions between these factors, are involved in the nosogenesis of MMD [8,9]. The interactions of genetic and environmental factors can change several metabolic pathways during the development of smoky blood vessels, thus altering the levels of metabolites in the body [10]. Therefore, the identification of specific metabolites can help elucidate the pathophysiology of this disease and explain the pathogenesis of this disease [11,12].

N-acetylneuraminic acid (Neu5Ac), one of the most common natural carbohydrates, is an essential building block of glycoproteins with many different biological functions, such as the synthesis of glycopeptides and glycolipids [13]. In addition, the biochemical derivatives of Neu5Ac are widely applied in drug synthesis and nutraceutical development [13]. Recent studies have shown that Neu5Ac has functions in promoting the inflammatory response of the immune system [13,14], improving insulin resistance [15], regulating lipoprotein metabolism [16], promoting thrombosis [17], and stimulating the proliferation and phenotypic transition of vascular smooth muscle cells [18]. However, to the best of our knowledge, varieties in serum Neu5Ac concentrations in patients with MMD and different subtypes, the relationship between serum Neu5Ac levels and angiogenesis in smoky blood vessels, and clinical prediction models have not been studied. Consequently, the purpose of our study was to investigate the action of serum Neu5Ac levels in a group of MMD patients.

In this paper, a case–control study was conducted to investigate the correlation between baseline serum Neu5Ac concentrations and the risk of MMD in Chinese patients. Additionally, regarding the clinical importance of MMD subtypes, we investigated the association of serum Neu5Ac concentrations with MMD subtypes at baseline. We speculated that multivariate analysis would be more valuable than univariate analysis for the early diagnosis of MMD. Therefore, we established a clinical prediction model to compare the efficiency of multivariate and univariate predictions of MMD and its subtypes. The calculation of the net reclassification index (NRI) and integrated discrimination improvement (IDI) showed that Neu5Ac is of great value in improving the accuracy of the model.

## 2. Materials and Methods

### 2.1. Study Design and Participants

In this study, we consecutively recruited 500 MMD patients from the Department of Neurosurgery, Beijing Tiantan Hospital, Capital Medical University, from 1 September 2020 to 31 December 2021. According to the guidelines published in Japan in 2012, MMD is diagnosed via digital subtraction angiography [19]: (1) unilateral or bilateral lesions, and (2) stenosis or occlusion of the terminal internal carotid and the proximal middle and anterior cerebral arteries. In this study, 82 pediatric patients, 11 adult patients over 60 years of age, and 47 patients with insufficient laboratory data were excluded. Finally, a total of 360 patients (258 with ischemic-type MMD and 102 with hemorrhagic-type MMD) were enrolled in this study (Figure 1). Eighty-nine healthy controls who received routine examinations served as the control group; the healthy controls were matched by age and sex with the MMD cases. None of these individuals or members of their close relatives had a history of MMD, according to the clinical interviews and their medical records. This study was approved by the Institutional Review Board of Beijing Tiantan Hospital (ethics number: KY2022-051-02) and was conducted in accordance with the Declaration of Helsinki of the World Medical Association published on the website of the Journal of American Medical Association. Informed consent was obtained from all patients or their representatives.

### 2.2. Data Collection

The clinical characteristics of the MMD patients acquired at admission included demographic data (age and sex), clinical features (heart rate, blood pressure, and body mass index), medical history (hypertension, diabetes mellitus, dyslipidemia, smoking status, and alcohol use), and initial clinical manifestations (ischemic type or hemorrhagic type). After the patients had rested for 15 min, systolic blood pressure (SBP) and diastolic blood pressure (DBP) of the right arm were measured using a standard mercury manometer in a seated position. At the same time, heart rate was recorded using an electrocardiogram. Body mass index (BMI) was calculated as weight (kg)/height (m^2^). Neurological status was assessed using the modified Rankin Scale (mRS) at admission and divided into two groups (0–2 and 3–5). The Suzuki stage of the MMD patients was defined based on the more severe side.

Fasting peripheral blood samples from all participants were used for routine and biochemical blood tests, as shown in Table 1. We also calculated peripheral inflammatory indicators based on laboratory examination data, which results are also presented in Table 1 [20,21]. For the measurement of Neu5Ac, blood samples were gathered from all patients after the diagnosis of MMD. The patients and healthy controls (HCs) were asked to fast for at least 12 h beforehand. After centrifugation, the blood samples were stored in a freezer at −80 °C until further analysis. Serum Neu5Ac concentrations in each patient and control subject were identified using liquid chromatography–tandem mass spectrometry, according to previous research [22,23]. We also detected *RNF213* p.R4810K variant with the primer designation of RNF213-4810F 5′-GCCCTCCATTTCTAGCACAC-3′ and RNF213-4810R 5′-AGCTGTGGCGAAAGCTTCTA-3′. 

### 2.3. Statistical Analyses

Statistical analyses were carried out using SPSS (version 26.0) and R software, version 4.2.3 (https://www.r-project.org) (accessed on 15 March 2023). We performed a complete case analysis for the enrolled participants. Categorical variables were expressed as frequencies, and continuous variables were described as mean with standard deviation (SD) or median with interquartile range (IQR). Continuous data were compared using *t*-tests or Mann–Whitney *U* tests between the two groups. Kruskal–Wallis tests or one-way ANOVA were used for comparisons among multiple groups. Categorical variables were compared with Pearson’s chi-squared tests, Fisher’s exact tests, and Kruskal–Wallis tests. Kendall correlation tests were used to assess the association between clinical characteristics and Neu5Ac quartiles.

We performed three logistic regression models to analyze the role of Neu5Ac (continuous and categorical variables) in MMD and its subtypes, including a Crude model, Model 1, and Model 2. Variables that were clinically meaningful and significantly different in the univariate analysis were considered for inclusion in the regression models (Appendix A). For the final model, collinearity analysis was performed with variance inflation factor (VIF) of less than 5 for all variables (Appendix A). The Crude model was the unadjusted regression model of Neu5Ac. Model 1 was adjusted for age, sex, heart rate, SBP, DBP, and BMI. Model 2 was further adjusted for platelet (PLT) count, alanine transaminase (ALT), alkaline phosphatase (ALP), glucose (Glu), urea, creatinine (Cr), triglyceride (TG), total cholesterol (TC), apolipoprotein B (ApoB), homocysteine (Hcy), platelet-to-neutrophil ratio (PNR), systemic immune-inflammation index (SII), and monocyte-to-HDL cholesterol ratio (MHR).

To compare the capacity of each regression model to discriminate MMD patients and subtypes of MMD patients from healthy controls, a logistic regression model was used to create receptor operating characteristic (ROC) curves for sensitivity and specificity, using the area under the ROC curve (AUC) to evaluate overall discrimination [24]. NRI and IDI were used to estimate the improvement in discrimination power using two different logistic regression models with 0.2, 0.4, 0.8, and 1 as the cutoff points [24,25], which are commonly utilized cutoff values for such analyses. *p* < 0.05 was considered statistical significance.

## 3. Results

### 3.1. Clinical and Laboratory Characteristics of the Study Participants

Overall, 360 adult patients with MMD and 89 healthy control subjects were included in our study. Baseline variables were compared between the HCs and MMD patients (Table 1). The median values of serum Neu5Ac concentrations in the MMD and control subjects were 4.62 μmol/L (interquartile range [IQR] of 1.85) and 4.33 μmol/L (IQR of 1.63), respectively (*p* = 0.001; Table 1; Figure 2A). Compared to the control subjects, MMD patients had higher levels of SBP, DBP, BMI, WBC count, NEUT count, ALT, ALP, TG, Hcy, SII, and MHR (*p* < 0.05 for all). Furthermore, MMD patients showed lower levels of TC, HDL-C, LDL-C, ApoA, and PNR (*p* < 0.05 for all). Patients with MMD had a higher tendency to have diabetes mellitus, hypertension, and hyperlipidemia and were more inclined to be smokers and drinkers than the control subjects (*p* < 0.05 for all).

The baseline characteristics of the MMD subtypes and healthy controls are summarized in Table 2, and the results are similar. The level of glucose in the ischemic-type MMD group was significantly higher than in the healthy control (HC) group (*p* = 0.015), indicating a higher risk of ischemia. Compared to the healthy control subjects, MMD patients in general (*p* = 0.001), ischemic-type MMD patients (*p* = 0.019), and hemorrhagic-type MMD patients (*p* < 0.001) had higher serum Neu5Ac concentrations. Furthermore, compared to ischemic-type MMD patients, hemorrhagic-type MMD patients had higher serum Neu5Ac concentrations (*p* < 0.05, Figure 2B).

### 3.2. Characteristics of MMD Patients with Different Neu5Ac Levels

Patients with MMD were categorized into two groups based on the median level of Neu5Ac: a low Neu5Ac group and a high Neu5Ac group. The clinical characteristics of the two groups are shown in Table 3. Although the difference was not statistically significant, patients in the high Neu5Ac group tended to have hemorrhagic-type MMD (*p* = 0.002). Furthermore, patients in the high Neu5Ac group had higher heart rates and higher levels of Cr, TC, LDL-C, and ApoB (*p* < 0.05 for all).

When all participants were stratified into quartiles based on their baseline serum Neu5Ac concentrations, participants with higher Neu5Ac levels tended to be hemorrhagic-type MMD patients and had higher levels of LY count, Cr, TC, LDL-C, and ApoB (*p* < 0.05 for all, Table 4). Although DBP was not statistically different between the groups, there was a positive linear correlation with Neu5Ac (*p* = 0.046).

### 3.3. Association of Neu5Ac Levels with Risk of MMD and MMD Subtypes

In this study, we observed a positive association between serum Neu5Ac and the risk of MMD in the Crude model (OR: 1.395, 95% CI: 1.141–1.706, *p* = 0.001; Table 5). In Model 1, the risk of MMD increases with each increment in Neu5Ac level (OR: 1.380, 95% CI: 1.125–1.694, *p* = 0.002). In Model 2, Neu5Ac also increases the risk of MMD (OR: 1.528, 95% CI: 1.178–1.983, *p* = 0.001). The ROC curves reveal that the predictive accuracy of Model 2 (AUC = 0.873; Figure 3A) improves significantly compared to the Crude model (AUC = 0.610) and Model 1 (AUC = 0.739). Similar results were found for the ROC curves of the three models for MMD subtypes (Table 5; Figure 3B,C).

We also compared the risk of MMD between patients with low and high Neu5Ac levels. We found that cases with high Neu5Ac levels had a significantly higher risk of MMD according to the Crude model (OR: 1.627, 95% CI: 1.016–2.606, *p* = 0.043), Model 1 (OR: 1.628, 95% CI: 0.988–2.681, *p* = 0.056), and Model 2 (OR: 1.882, 95% CI: 1.037–3.413, *p* = 0.037; Table 5). However, for Models 1 and 2 of the ischemic MMD group, the OR of high Neu5Ac level versus low Neu5Ac level is less than 1 and *p* > 0.05. For the hemorrhagic MMD group, the Crude model (OR: 3.000, 95% CI: 1.662–5.414, *p* < 0.001), Model 1 (OR: 3.420, 95% CI: 1.823–6.416, *p* < 0.001), and Model 2 (OR: 3.154, 95% CI: 1.472–6.757, *p* = 0.003) show better adaptability. The AUC values of MMD overall show an improvement in the Crude model (AUC = 0.560; Figure 4A), Model 1 (AUC = 0.728), and Model 2 (AUC = 0.863). Similar results were observed in the Crude model, Model 1, and Model 2 of ischemic and hemorrhagic MMD (Figure 4B,C).

Furthermore, we evaluated Neu5Ac as quartiles and analyzed its risk in MMD (Figure 5A, Appendix A). In contrast to the cases in Q1 of Neu5Ac, the cases in Q4 of Neu5Ac had a significantly higher risk of MMD according to the Crude model (Q4, OR: 3.189, 95% CI: 1.540–6.604, *p* = 0.002, *p* for trend = 0.004), Model 1 (Q4, OR: 2.884, 95% CI: 1.350–6.159, *p* = 0.006, *p* for trend = 0.010), and Model 2 (Q4, OR: 4.039, 95% CI: 1.650–9.888, *p* = 0.002, *p* for trend = 0.004). The AUC values of the ROC curves increase with the adjustment for cofounders in the models (Crude model, AUC = 0.597; Model 1, AUC = 0.733; and Model 2, AUC = 0.869) (Figure 6A). We found that the proportion of ischemic and hemorrhagic MMD events increased with an increase in the Neu5Ac quartiles (Figure 5B,C; Appendix A). Compared to the lowest quartile (Q1), the cases in the fourth (Q4) Neu5Ac quartile show a significant correlation with ischemic MMD according to the Crude model (Q4, OR: 2.377, 95% CI: 1.180–4.790, *p* = 0.015, *p* for trend = 0.012). In Model 2, compared to Q1 of serum Neu5Ac level, the adjusted OR (95% CI, *p* value) for ischemic MMD in Q2 and Q4 is 2.493 (1.018–6.107, *p* = 0.046, *p* for trend = 0.026) and 3.123 (1.232–7.918, *p* = 0.016), respectively. The AUCs of the ROC curves in the Crude model, Model 1, and Model 2 are 0.587, 0.754, and 0.893, respectively (Figure 6B). Moreover, the third (Q3) and fourth (Q4) Neu5Ac quartiles are significantly correlated with the risk of hemorrhagic MMD in the Crude model (Q3, OR: 2.712, 95% CI: 1.180–6.237, *p* = 0.019; Q4, OR: 5.812, 95% CI; 2.389–14.144, *p* < 0.001, *p* for trend < 0.001) and Model 1 (Q3, OR: 2.960, 95% CI: 1.220–7.178, *p* = 0.016; Q4, OR: 6.598, 95% CI: 2.588–16.819, *p* < 0.001, *p* for trend < 0.001). In Model 2, compared to Q1 of serum Neu5Ac level, the adjusted OR (95% CI, *p* value) for hemorrhagic MMD in Q4 is 8.442 (2.703–26.364, *p* < 0.001, *p* for trend < 0.001). The AUCs of the ROC curves in the Crude model, Model 1, and Model 2 are 0.670, 0.733, and 0.867, respectively (Figure 6C).

### 3.4. Predictive Capacity of Neu5Ac in Clinical Risk Models

To explore the predictive value of serum Neu5Ac, we calculated categorical NRI and IDI for models containing serum Neu5Ac concentrations, Neu5Ac binary outcomes, and Neu5Ac quartiles on top of our previously established clinical risk model (Table 6). The basic model was adjusted for age, sex, SBP, DBP, BMI, PLT count, ALT, ALP, Glu, urea, Cr, TG, TC, ApoB, Hcy, PNR, SII, and MHR. The basic model plus serum Neu5Ac concentrations for MMD has a categorical NRI of 12.4% (95% CI: 3.3–21.4%; *p* = 0.007) and an IDI of 2.8% (95% CI: 0.8–4.7%; *p* = 0.005); the model for ischemic MMD has an IDI of 1.8% (95% CI: 0.1–3.5%; *p* = 0.042); and the model for hemorrhagic MMD has a categorical NRI of 19.9% (95% CI: 7.4–32.4%; *p* = 0.002) and an IDI of 6.5% (95% CI: 3.0–10.0%; *p* < 0.001). In addition, the basic model plus Neu5Ac binary outcomes for MMD has a categorical NRI of 7.0% (95% CI: 0.1–14.0%; *p* = 0.047), and the model for hemorrhagic MMD has a categorical NRI of 15.2% (95% CI: 2.9–27.4%; *p* = 0.015) and an IDI of 3.4% (95% CI: 0.7–6.1%; *p* = 0.014). Moreover, the basic model plus Neu5Ac quartiles for MMD has an IDI of 2.0% (95% CI: 0.3–3.7%; *p* = 0.024), and the model for hemorrhagic MMD has a categorical NRI of 16.6% (95% CI: 3.3–29.8%; *p* = 0.014) and an IDI of 5.6% (95% CI: 2.2–8.9%; *p* = 0.001).

## 4. Discussion

In this study, we investigated the association between baseline serum Neu5Ac and the risk of MMD in a Chinese population. Then, we compared the difference in clinical characteristics between HCs, MMD patients, and MMD subtypes. Our findings revealed that serum Neu5Ac levels were significantly higher in the MMD group and its subtypes than those in the HC group. We expounded and proved that higher Neu5Ac levels were correlated with an increased risk of MMD, especially of hemorrhagic MMD, even after adjusting for laboratory results and other basic clinical characteristics. Moreover, we created a clinical diagnostic model and examined its predictive accuracy using the ROC curve. Including Neu5Ac in the model offered substantial improvement in the risk reclassification and discrimination of MMD and its subtypes. In medical and clinical settings, serum Neu5Ac can be used as a novel biomarker for the diagnosis of MMD, and targeting Neu5Ac may serve as a novel therapeutic strategy for MMD. Furthermore, the multifactorial clinical prediction model including serum Neu5Ac concentrations developed in this study could be used to predict MMD and its subtypes. To the best of our knowledge, this is the largest and first study to date assessing the relationship between serum Neu5Ac levels and the risk of MMD.

In recent years, the role of body metabolites in vascular diseases has attracted extensive attention and interest. Overall quantitative and qualitative analyses of multifarious small molecules in patients with coronary artery disease and other cardiovascular diseases have revealed changes in biological endogenous metabolites after their measurement in the internal and external environments, which has been shown to help identify disease-related disorders. A panel of metabolic markers can help facilitate the early detection and treatment of diseases [26,27,28]. Neu5Ac, known as sialic acid, is a naturally occurring glucosamine. It was originally isolated from bovine submandibular mucin, which belongs to a family of monosaccharides with a backbone consisting of nine carbon atoms and is structurally diverse. Recently, it has been argued that Neu5Ac has diversified biological functions. As a viral receptor, it is closely associated with tumor transformation, cancer metastasis, infiltration, loss of contact inhibition, decreased cell adhesion, and antigenicity [29,30,31]. According to previous studies, serum Neu5Ac is linked to atrial fibrillation. High serum levels of Neu5Ac may lead to left atrial enlargement and fibrosis, which facilitates the occurrence of atrial fibrillation [32]. In one study, patients with heart failure had elevated plasma levels of Neu5Ac compared to patients without heart failure. Elevated serum Neu5Ac levels predicted a poor long-term prognosis in patients with heart failure independent of traditional risk factors [33]. Another study showed that serum Neu5Ac was related to myocardial injury in patients with acute coronary syndrome. This explained the severity of coronary artery disease and predicted poor outcomes in these patients [34]. In conclusion, serum Neu5Ac, as a metabolite, is a significant risk factor for cardiovascular disease and suggests a harmful prognosis when its concentration is elevated. Nevertheless, Neu5Ac is less studied in cerebrovascular diseases.

MMD is generally identified by stenosis at the end of the internal carotid arteries (ICAs) and the consequent dilatation of compensatory blood vessels. Histologically, it is featured by the thickening and corrugation of the elastic lamina, hypertrophy of intimal fibroblasts, and hyperplasia of smooth muscle cells involving the ICAs and proximal anterior and middle cerebral arteries [35]. Zhang et al. [22] used untargeted metabolomics to analyze a large number of plasma samples and found that Neu5Ac can bind to RhoA and Cdc42 to activate the Rho/ROCK signaling pathway. There are two isoforms of Rho kinase, Rock1 and Rock2, which are, respectively, expressed in vascular smooth muscle and the heart. Activation of the Rho/Rho kinase signaling pathway through multiple pathways brings about myosin light chain (MLC) phosphorylation and integrin aggregation, thereby increasing endothelial cell permeability. The effects of upregulation of Neu5Ac and activation of the Rho/Rho kinase signaling pathway may contain monocyte/macrophage migration, transport of oxidized low-density lipoprotein, endothelial dysfunction, and phenotypic transition and proliferation of vascular smooth muscle cells, which may be involved in the process of terminal internal carotid artery stenosis. At present, it is believed that an increase in serum Neu5Ac concentrations may boost the formation of atherosclerosis by increasing the inflammatory response, destroying the body’s lipid metabolism, and promoting platelet thrombosis [36,37]. For a proportion of patients with MMD, atherosclerosis of cerebral blood vessels is the cause of stenosis.

Neu5Ac may also play a role in angiogenesis and dilatation of compensatory blood vessels in patients with MMD. During embryogenesis and wound healing, the formation of new blood vessels from existing blood vessels is called angiogenesis. Angiogenesis is stimulated by angiogenic growth factors (AGFs) released by inflammatory cells. Among them, the most crucial AGFs are the vascular endothelial growth factor (VEGF) family, including VEGF-A, B, C, D, E, and the placental growth factor [38]. The interaction between VEGFs and polysialic acid plays a significant role in the process of angiogenesis [39]. In a hypoxic environment, it is easy for the body to generate its own compensatory angiogenesis, and Neu5Ac may promote this process. Saliva’s liquefaction status can affect growth factor–receptor interactions and associated signaling during angiogenesis [40]. Ganglioside is a sialyl-glycosphingolipid that is incorporated into endothelial cell membranes and increases cellular responsiveness to AGFs [41,42]. N-glycans at the α2,6-salicylic acid terminus of vascular receptors, such as VEGFR2, are required for VEGF binding and pro-angiogenic activation of endothelial cells [43,44]. Salivation of α2,6-salicylic acid mediates the interaction of cognate platelet endothelial cell adhesion molecules (PECAM). PECAM salivary secretion interacts with two other salivary receptors, VEGFR2 and integrin β3, and inhibition of salivary secretion in ST6GAL1-/- mice prevents the interaction at the endothelial surface, induces endothelial cell apoptosis, and inhibits angiogenesis. In conclusion, the elevation of Neu5Ac may have effects on terminal ICA stenosis and moyamoya angiogenesis [45].

Although our study has potential benefits as the first step to illuminate the effect of Neu5Ac in MMD, our study also has limitations. First, we found that Neu5Ac is related to the risk of MMD in the Chinese population, but further verification is necessary for the broader Asian population. Second, Neu5Ac was measured at only one time point, and no dietary history was taken to estimate the effect of diet on Neu5Ac levels in the body. Third, before blood collection, no information was collected on antibiotic use or gastrointestinal symptoms. Fourth, we chose a case–control study design that considered cost and efficiency; to confirm our results, it needs to be corroborated in a larger prospective cohort. Fifth, our study included only MMD patients and did not address moyamoya syndrome or other cerebrovascular diseases. Finally, despite the adjustment for known factors, there are other traditional potential confounders that we did not use to compare the cases and controls. Future studies are required to identify the molecular mechanism that mediates the effect of Neu5Ac on the etiology of MMD and to determine whether Neu5Ac is the potential therapeutic target or the only biomarker for MMD.

## 5. Conclusions

Taken together, our results suggest that the Neu5Ac–MMD risk association we observed may be independent. Therefore, serum Neu5Ac can be a new biomarker for diagnosing MMD, and targeting Neu5Ac in the future may be a potential therapeutic strategy for MMD. Additionally, we created a multifactorial clinical prediction model including serum Neu5Ac concentrations to predict MMD and its subtypes. Nevertheless, our results require well-designed basic and clinical trials to further explore the association of higher Neu5Ac levels with the pathogenesis of MMD.

## Figures and Tables

**Figure 1 brainsci-13-00913-f001:**
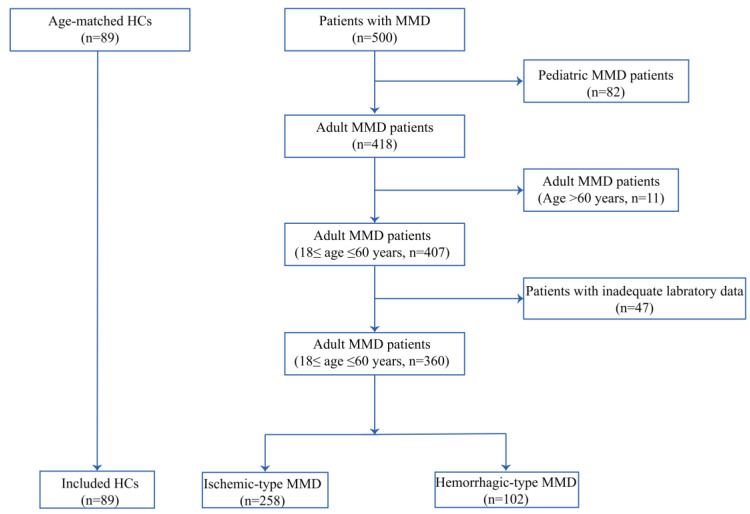
Flow chart of the study participants. HCs, healthy controls; MMD, moyamoya disease.

**Figure 2 brainsci-13-00913-f002:**
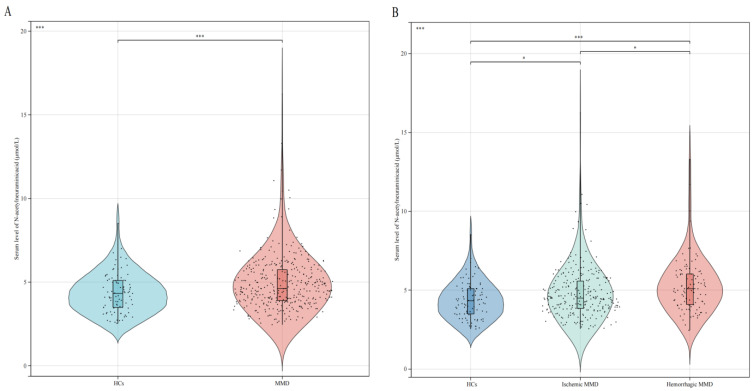
Levels of Neu5Ac between HCs and MMD subtypes: (**A**) comparison of Neu5Ac levels between HCs and MMD patients, and (**B**) comparison of Neu5Ac levels between HCs and MMD subtypes. HCs, healthy controls; MMD, moyamoya disease. * *p* < 0.05, *** *p* ≤ 0.001.

**Figure 3 brainsci-13-00913-f003:**
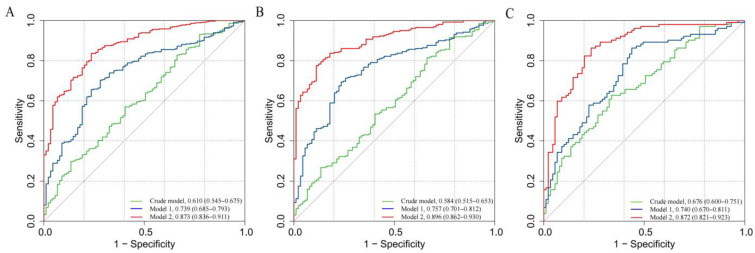
ROC curves of Neu5Ac in different models for the risk of MMD and its subtypes: (**A**) MMD overall; (**B**) ischemic-type MMD; and (**C**) hemorrhagic-type MMD.

**Figure 4 brainsci-13-00913-f004:**
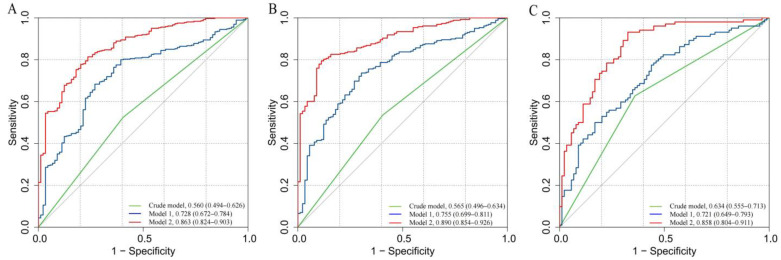
ROC curves of low and high levels of Neu5Ac in different models for the risk of MMD and its subtypes: (**A**) MMD overall; (**B**) ischemic-type MMD; and (**C**) hemorrhagic-type MMD.

**Figure 5 brainsci-13-00913-f005:**
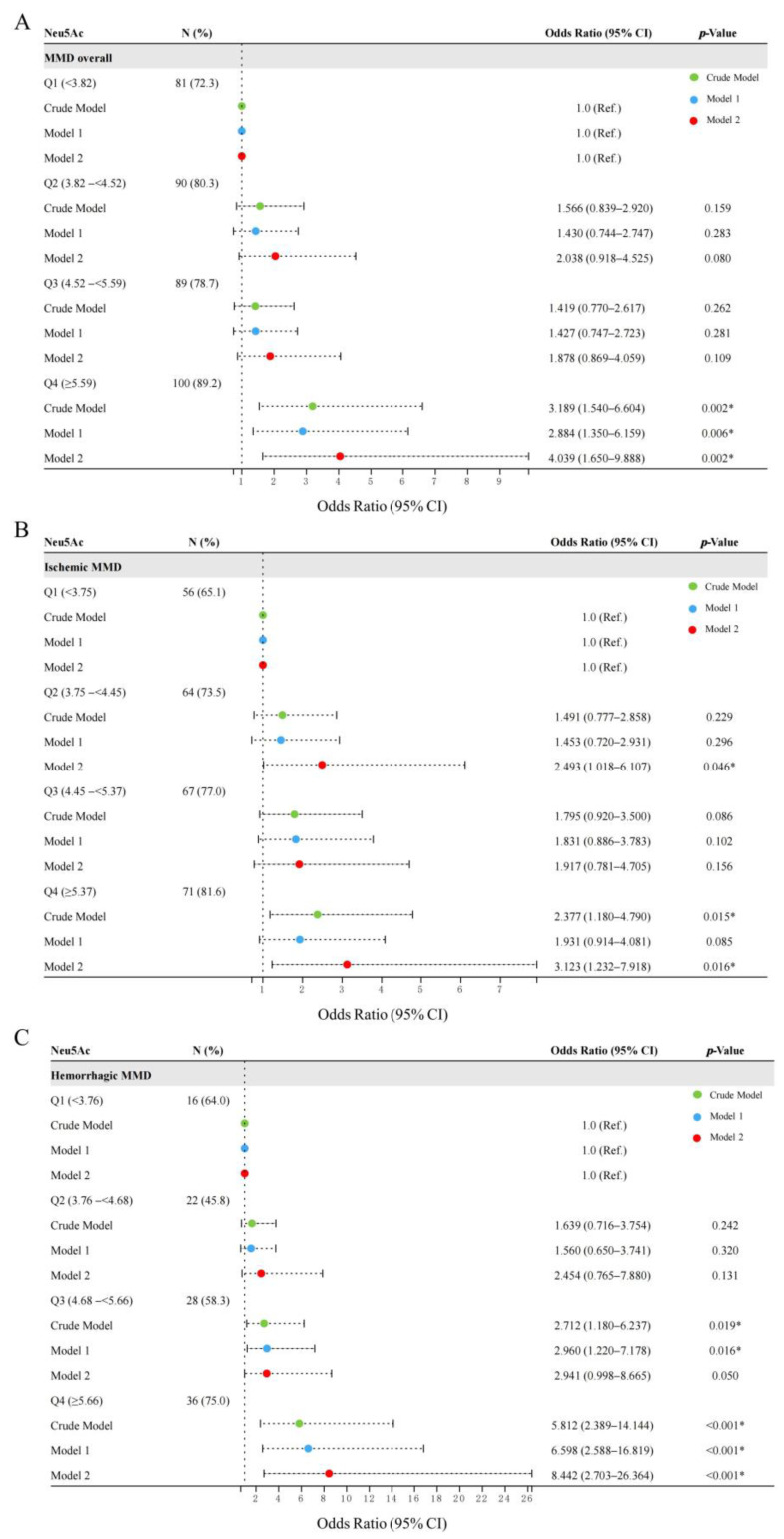
Forest plots for the association of Neu5Ac with the risk of MMD and its subtypes: (**A**) MMD overall; (**B**) ischemic-type MMD; and (**C**) hemorrhagic-type MMD. * *p* < 0.05 indicates significant difference.

**Figure 6 brainsci-13-00913-f006:**
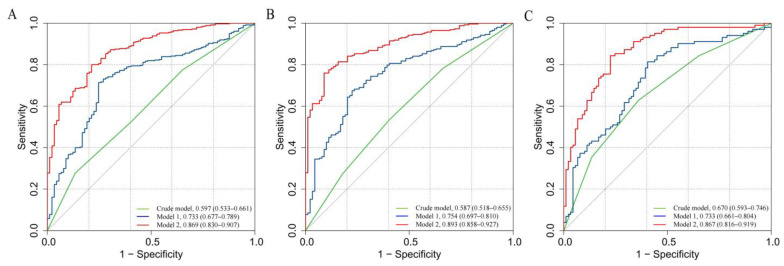
ROC curves of Neu5Ac quartiles in different models for the risk of MMD and its subtypes: (**A**) MMD overall; (**B**) ischemic-type MMD; and (**C**) hemorrhagic-type MMD.

**Table 1 brainsci-13-00913-t001:** Comparison of baseline characteristics between the HC and MMD groups.

Variables	Healthy Controls (*n* = 89)	MMD Patients (*n* = 360)	*p*-Value
Age (years), mean ± SD	39.81 ± 11.57	41.54 ± 10.32	0.200
Gender (female), *n* (%)	52 (58.4)	210 (58.3)	0.987
Clinical features, mean ± SD			
Heart rate, bpm	77.79 ± 9.73	78.54 ± 6.42	0.486
SBP, mmHg	123.64 ± 11.77	132.34 ± 12.83	<0.001 *
DBP, mmHg	78.46 ± 8.35	81.84 ± 9.32	0.002 *
BMI, kg/m^2^	23.96 ± 3.39	25.45 ± 4.52	0.004 *
History of risk factors, *n* (%)			
Current smoking	2 (2.2)	71 (19.7)	<0.001 *
Current alcohol abuse	0 (0.0)	42 (11.7)	0.001 *
Diabetes mellitus	0 (0.0)	59 (16.4)	<0.001 *
Hypertension	0 (0.0)	131 (36.4)	<0.001 *
Hyperlipidemia	0 (0.0)	54 (15.0)	<0.001 *
Laboratory results, median (IQR)			
WBC count, 10^9^/L	6.03 (1.88)	6.81 (2.48)	<0.001 *
LY count, 10^9^/L	1.91 (0.71)	1.92 (0.87)	0.247
NEUT count, 10^9^/L	3.44 (1.62)	4.21 (1.88)	<0.001 *
MONO count, 10^9^/L	0.35 (0.14)	0.35 (0.17)	0.295
PLT count, 10^9^/L	233.00 (87.00)	248.00 (75.75)	0.290
ALT, U/L	18.40 (10.05)	21.05 (17.58)	0.045 *
ALP, U/L	57.00 (21.50)	69.30 (26.88)	<0.001 *
Glu, mmol/L	5.04 (0.62)	5.11 (1.07)	0.237
Urea, mmol/L	4.70 (1.85)	4.55 (1.80)	0.208
Cr, μmol/L	57.7 (19.20)	54.65 (20.65)	0.146
TG, mmol/L	0.87 (0.62)	1.20 (0.80)	<0.001 *
TC, mmol/L	4.62 (0.98)	4.23 (1.28)	<0.001 *
HDL-C, mmol/L	1.53 (0.41)	1.30 (0.37)	<0.001 *
LDL-C, mmol/L	2.69 (0.87)	2.40 (1.13)	<0.001 *
ApoA, g/L	1.39 (0.28)	1.29 (0.30)	<0.001 *
ApoB, g/L	0.77 (0.27)	0.82 (0.27)	0.318
Hcy, μmol/L	10.62 (3.97)	12.00 (5.78)	0.001 *
PNR	67.42 (37.22)	60.17 (31.13)	0.001 *
SII	414.33 (289.40)	536.13 (383.84)	<0.001 *
MHR	0.23 (0.11)	0.28 (0.18)	<0.001 *
Neu5Ac, μmol/L	4.33 (1.63)	4.62 (1.85)	0.001 *

HC, healthy control; MMD, moyamoya disease; SD, standard deviation; SBP, systolic blood pressure; DBP, diastolic blood pressure; BMI, body mass index; IQR, interquartile range; WBC, white blood cell; LY, lymphocyte; NEUT, neutrophil; MONO, monocyte; PLT, platelet; ALT, alanine transaminase; ALP, alkaline phosphatase; Glu, glucose; Cr, creatinine; TG, triglyceride; TC, total cholesterol; HDL-C, high-density lipoprotein cholesterol; LDL-C, low-density lipoprotein cholesterol; ApoA, apolipoprotein A; ApoB, apolipoprotein B; Hcy, homocysteine; PNR, platelet-to-neutrophil ratio; SII, systemic immune-inflammation index; MHR, monocyte-to-HDL cholesterol ratio; Neu5Ac, N-acetylneuraminic acid. * *p* < 0.05 indicates significant difference.

**Table 2 brainsci-13-00913-t002:** Comparison of baseline characteristics between HCs and MMD subtypes.

Variables	Health Controls (*n* = 89)	Ischemic MMD (*n* = 258)	*p*-Value	Hemorrhagic MMD (*n* = 102)	*p*-Value
Age (years) mean ± SD	39.81 ± 11.57	41.53 ± 10.14	0.214	41.55 ± 10.82	0.284
Gender (female), *n* (%)	52 (58.4)	142 (55.0)	0.621	68 (66.7)	0.294
Clinical features, mean ± SD					
Heart rate, bpm	77.79 ± 9.73	78.25 ± 6.60	0.678	79.29 ± 5.89	0.205
SBP, mmHg	123.64 ± 11.77	133.48 ± 12.78	<0.001 *	129.45 ± 12.54	0.001 *
DBP, mmHg	78.46 ± 8.35	82.39 ± 9.39	0.001 *	80.46 ± 9.03	0.116
BMI, kg/m^2^	23.96 ± 3.39	25.93 ± 4.60	<0.001 *	24.27 ± 4.12	0.581
History of risk factors, *n* (%)					
Current smoking	2 (2.2)	54 (20.9)	<0.001 *	17 (16.7)	0.001 *
Current alcohol abuse	0 (0.0)	34 (13.2)	<0.001 *	8 (7.8)	0.008 *
Diabetes mellitus	0 (0.0)	55 (21.3)	<0.001 *	4 (3.9)	0.125
Hypertension	0 (0.0)	101 (39.1)	<0.001 *	30 (29.4)	<0.001 *
Hyperlipidemia	0 (0.0)	45 (17.4)	<0.001 *	9 (8.8)	0.004 *
Laboratory results, median (IQR)					
WBC count, 10^9^/L	6.03 (1.88)	7.00 (2.42)	<0.001 *	6.43 (2.48)	0.021 *
LY count, 10^9^/L	1.91 (0.71)	2.04 (0.87)	0.021 *	1.70 (0.83)	0.106
NEUT count, 10^9^/L	3.44 (1.62)	4.31 (1.83)	0.158	3.88 (1.85)	0.004 *
MONO count, 10^9^/L	0.35 (0.14)	0.36 (0.17)	<0.001 *	0.35 (0.17)	0.991
PLT count, 10^9^/L	233.00 (87.00)	249.50 (71.75)	0.156	243.50 (75.75)	0.993
ALT, U/L	18.40 (10.05)	21.90 (19.90)	0.014 *	18.80 (15.20)	0.611
ALP, U/L	57.00 (21.50)	68.80 (26.75)	<0.001 *	69.45 (26.35)	<0.001 *
Glu, mmol/L	5.04 (0.62)	5.17 (1.14)	0.015 *	4.91 (0.68)	0.068
Urea, mmol/L	4.70 (1.85)	4.50 (1.80)	0.177	4.60 (2.03)	0.465
Cr, μmol/L	57.7 (19.20)	55.65 (21.05)	0.279	52.95 (21.63)	0.063
TG, mmol/L	0.87 (0.62)	1.22 (0.79)	<0.001 *	1.13 (0.82)	0.012 *
TC, mmol/L	4.62 (0.98)	4.10 (1.33)	<0.001 *	4.35 (1.16)	0.170
HDL-C, mmol/L	1.53 (0.41)	1.27 (0.38)	<0.001 *	1.34 (0.35)	0.001 *
LDL-C, mmol/L	2.69 (0.87)	2.24 (1.14)	<0.001 *	2.55 (1.06)	0.716
ApoA, g/L	1.39 (0.28)	1.29 (0.31)	<0.001 *	1.30 (0.28)	0.004 *
ApoB, g/L	0.77 (0.27)	0.82 (0.27)	0.692	0.83 (0.30)	0.043 *
Hcy, μmol/L	10.62 (3.97)	12.03 (6.33)	0.001 *	11.95 (4.94)	0.005 *
PNR	67.42 (37.22)	60.17 (28.09)	<0.001 *	60.29 (39.24)	0.016 *
SII	414.33 (289.40)	532.68 (350.76)	0.001 *	541.80 (444.31)	0.003 *
MHR	0.23 (0.11)	0.29 (0.18)	<0.001 *	0.24 (0.16)	0.088
Neu5Ac, μmol/L	4.33 (1.63)	4.50 (1.76)	0.019 *	5.09 (1.97)	<0.001 *

HCs, healthy controls; MMD, moyamoya disease; SD, standard deviation; SBP, systolic blood pressure; DBP, diastolic blood pressure; BMI, body mass index; IQR, interquartile range; WBC, white blood cell; LY, lymphocyte; NEUT, neutrophil; MONO, monocyte; PLT, platelet; ALT, alanine transaminase; ALP, alkaline phosphatase; Glu, glucose; Cr, creatinine; TG, triglyceride; TC, total cholesterol; HDL-C, high-density lipoprotein cholesterol; LDL-C, low-density lipoprotein cholesterol; ApoA, apolipoprotein A; ApoB, apolipoprotein B; Hcy, homocysteine; PNR, platelet-to-neutrophil ratio; SII, systemic immune-inflammation index; MHR, monocyte-to-HDL cholesterol ratio; Neu5Ac, N-acetylneuraminic acid. * *p* < 0.05 indicates significant difference.

**Table 3 brainsci-13-00913-t003:** Baseline characteristics of MMD patients with low and high Neu5Ac levels.

Variables	Low Neu5Ac (2.45–4.62, *n* = 180)	High Neu5Ac (4.62–16.25, *n* = 180)	*p*-Value
Age (years), mean ± SD	41.38 ± 10.35	41.69 ± 10.32	0.771
Gender (female), *n* (%)	113 (62.8)	97 (53.9)	0.109
Clinical features, mean ± SD			
Heart rate, bpm	77.8 ± 5.68	79.29 ± 7.02	0.028 *
SBP, mmHg	131.77 ± 13.27	132.9 ± 12.38	0.405
DBP, mmHg	80.98 ± 9.22	82.71 ± 9.36	0.078
BMI, kg/m^2^	25.27 ± 4.77	25.64 ± 4.27	0.431
*RNF213* p.R4810K, *n* (%)			
Wild type	145 (81.9)	141 (81.5)	0.919
Mutant	32 (18.1)	32 (18.5)	
History of risk factors, *n* (%)			
Current smoking	30 (16.7)	41 (22.8)	0.185
Current alcohol abuse	15 (8.3)	27 (15.0)	0.070
Diabetes mellitus	29 (16.1)	30 (16.7)	0.887
Hypertension	62 (34.4)	69 (38.3)	0.511
Hyperlipidemia	29 (16.1)	25 (13.9)	0.658
Laboratory results, median (IQR)			
WBC count, 10^9^/L	6.85 (2.54)	6.81 (2.43)	0.917
LY count, 10^9^/L	1.88 (0.88)	1.99 (0.85)	0.061
NEUT count, 10^9^/L	4.29 (1.80)	4.19 (1.94)	0.468
MONO count, 10^9^/L	0.34 (0.16)	0.37 (0.17)	0.237
PLT count, 10^9^/L	247.50 (74.25)	248.50 (85.25)	0.433
ALT, U/L	21.15 (18.08)	21.00 (18.03)	0.687
ALP, U/L	68.30 (28.38)	69.95 (24.72)	0.721
Glu, mmol/L	5.13 (1.14)	5.06 (0.98)	0.979
Urea, mmol/L	4.45 (1.88)	4.60 (1.95)	0.425
Cr, μmol/L	53.55 (18.80)	56.75 (22.08)	0.047 *
TG, mmol/L	1.20 (0.74)	1.19 (0.83)	0.936
TC, mmol/L	4.03 (1.30)	4.26 (1.22)	0.040 *
HDL-C, mmol/L	1.31 (0.39)	1.29 (0.33)	0.495
LDL-C, mmol/L	2.24 (1.11)	2.43 (1.23)	0.018 *
ApoA, g/L	1.30 (0.30)	1.28 (0.31)	0.308
ApoB, g/L	0.79 (0.27)	0.86 (0.28)	0.017 *
Hcy, μmol/L	12.05 (6.32)	12.00 (5.38)	0.919
PNR	58.90 (28.77)	60.80 (36.26)	0.213
SII	544.14 (434.77)	525.64 (344.73)	0.219
MHR	0.26 (0.18)	0.29 (0.18)	0.376
Neu5Ac, μmol/L	3.90 (0.83)	5.74 (1.27)	<0.001 *
Clinical type, *n* (%)			0.002 *
Ischemic type	142 (78.9)	116 (64.4)	
Hemorrhagic type	38 (21.1)	64 (35.6)	
Admission mRS, *n* (%)			0.859
0–2	163 (90.6)	162 (90.0)	
3–5	17 (9.4)	18 (10.0)	
Suzuki stage, *n* (%)			0.754
1–2	48 (26.7)	52 (28.9)	
3–4	95 (52.8)	88 (48.9)	
5–6	37 (20.6)	40 (22.2)	

MMD, moyamoya disease; Neu5Ac, N-acetylneuraminic acid; SD, standard deviation; SBP, systolic blood pressure; DBP, diastolic blood pressure; BMI, body mass index; IQR, interquartile range; WBC, white blood cell; LY, lymphocyte; NEUT, neutrophil; MONO, monocyte; PLT, platelet; ALT, alanine transaminase; ALP, alkaline phosphatase; Glu, glucose; Cr, creatinine; TG, triglyceride; TC, total cholesterol; HDL-C, high-density lipoprotein cholesterol; LDL-C, low-density lipoprotein cholesterol; ApoA, apolipoprotein A; ApoB, apolipoprotein B; Hcy, homocysteine; PNR, platelet-to-neutrophil ratio; SII, systemic immune-inflammation index; MHR, monocyte-to-HDL cholesterol ratio; mRS, modified Rankin Scale. * *p* < 0.05 indicates significant difference.

**Table 4 brainsci-13-00913-t004:** Baseline characteristics of MMD patients according to the quartiles of Neu5Ac.

Variables	Neu5Ac				*p*-Value	Kendall’s Tau-b Coefficient	*p*-Value
	Q1 (2.45–3.89, *n* = 90)	Q2 (3.89–4.62, *n* = 90)	Q3 (4.62–5.75, *n* = 90)	Q4 (5.75–16.25, *n* = 90)			
Age (years), mean ± SD	40.32 ± 9.94	42.43 ± 10.70	41.82 ± 11.24	41.57 ± 9.37	0.522	0.022	0.579
Gender (female), *n* (%)	55 (61.1)	58 (64.4)	49 (54.4)	48 (53.3)	0.379	0.069	0.152
Clinical features, mean ± SD							
Heart rate, bpm	77.51 ± 5.75	78.09 ± 5.63	79.64 ± 7.56	78.93 ± 6.45	0.054	0.070	0.090
SBP, mmHg	130.78 ± 12.29	132.77 ± 14.17	133.43 ± 12.41	132.37 ± 12.40	0.370	0.044	0.277
DBP, mmHg	80.78 ± 8.97	81.18 ± 9.50	82.47 ± 9.60	82.96 ± 9.16	0.076	0.081	0.046 *
BMI, kg/m^2^	24.55 ± 4.18	25.98 ± 5.21	25.38 ± 4.41	25.91 ± 4.14	0.104	0.076	0.055
RNF213 p.R4810K, *n* (%)					0.477	−0.020	0.682
Wild type	68 (77.30)	77 (86.5)	71 (81.6)	70 (81.4)			
Mutant	20 (22.7)	12 (13.5)	16 (18.4)	16 (18.6)			
History of risk factors, *n* (%)							
Current smoking	18 (20.0)	12 (13.3)	19 (21.1)	22 (24.4)	0.305	0.054	0.261
Current alcohol abuse	8 (8.9)	7 (7.8)	12 (13.3)	15 (16.7)	0.252	0.092	0.057
Diabetes mellitus	13 (14.4)	16 (17.8)	12 (13.3)	18 (20.0)	0.623	0.034	0.484
Hypertension	26 (28.9)	36 (40.0)	37 (41.1)	32 (35.6)	0.323	0.045	0.353
Hyperlipidemia	14 (15.6)	15 (16.7)	15 (16.7)	10 (11.1)	0.740	−0.038	0.429
Laboratory results, median (IQR)							
WBC count, 10^9^/L	6.92 (2.46)	6.82 (2.52)	6.65 (2.24)	6.90 (2.57)	0.953	0.016	0.685
LY count, 10^9^/L	1.87 (0.91)	1.88 (0.84)	1.98 (0.76)	2.03 (0.93)	0.010 *	0.083	0.036 *
NEUT count, 10^9^/L	4.29 (1.72)	4.28 (1.84)	4.18 (1.90)	4.22 (2.10)	0.393	−0.018	0.654
MONO count, 10^9^/L	0.35 (0.20)	0.33 (0.14)	0.37 (0.17)	0.36 (0.14)	0.509	0.013	0.752
PLT count, 10^9^/L	241.00 (74.50)	251.00 (66.50)	245.50 (73.50)	254.50 (91.00)	0.081	0.061	0.124
ALT, U/L	22.10 (21.60)	20.80 (14.80)	19.25 (18.80)	21.55 (17.18)	0.669	−0.009	0.816
ALP, U/L	65.70 (27.30)	69.70 (28.80)	70.30 (31.93)	68.85 (23.15)	0.577	0.010	0.791
Glu, mmol/L	5.06 (0.90)	5.19 (1.19)	4.95 (0.85)	5.17 (1.08)	0.511	0.055	0.161
Urea, mmol/L	4.40 (2.05)	4.60 (1.48)	4.60 (1.85)	4.60 (2.10)	0.330	0.027	0.492
Cr, μmol/L	54.25 (19.15)	53.20 (18.73)	54.15 (19.90)	60.50 (24.25)	0.007 *	0.092	0.019 *
TG, mmol/L	1.16 (0.77)	1.22 (0.76)	1.22 (1.02)	1.16 (0.74)	0.292	0.003	0.935
TC, mmol/L	3.92 (1.12)	4.13 (1.42)	4.23 (1.38)	4.26 (0.95)	0.021 *	0.092	0.019 *
HDL-C, mmol/L	1.29 (0.41)	1.34 (0.36)	1.33 (0.37)	1.26 (0.29)	0.642	−0.033	0.407
LDL-C, mmol/L	2.18 (1.11)	2.33 (1.14)	2.40 (1.35)	2.51 (0.99)	0.016 *	0.108	0.006 *
ApoA, g/L	1.28 (0.27)	1.32 (0.36)	1.30 (0.31)	1.23 (0.27)	0.409	−0.036	0.359
ApoB, g/L	0.76 (0.27)	0.82 (0.27)	0.85 (0.29)	0.87 (0.27)	0.007 *	0.118	0.003 *
Hcy, μmol/L	12.00 (5.79)	12.05 (6.21)	11.65 (5.57)	12.26 (5.44)	0.877	0.028	0.480
PNR	56.90 (27.30)	59.49 (30.08)	60.45 (32.95)	61.38 (38.77)	0.136	0.055	0.167
SII	526.35 (457.77)	573.51 (426.91)	507.85 (303.91)	532.61 (382.66)	0.309	−0.039	0.324
MHR	0.29 (0.19)	0.25 (0.15)	0.28 (0.18)	0.29 (0.17)	0.699	0.011	0.771
Neu5Ac, μmol/L	3.43 (0.64)	4.26 (0.39)	5.09 (0.49)	6.35 (1.08)	<0.001 *	0.867	<0.001 *
Clinical type, *n* (%)					0.024 *	0.131	0.007 *
Ischemic type	70 (77.8)	72 (80)	59 (65.6)	57 (63.3)			
Hemorrhagic type	20 (22.2)	18 (20.0)	31 (34.4)	33 (36.7)			
Admission mRS, *n* (%)					0.160	0.034	0.475
0–2	86 (95.6)	77 (85.6)	80 (88.9)	82 (91.1)			
3–5	4 (4.4)	13 (14.4)	10 (11.1)	8 (8.9)			
Suzuki stage, *n* (%)					0.666	−0.006	0.898
1–2	20 (22.2)	28 (31.1)	30 (33.3)	22 (24.4)			
3–4	50 (55.6)	45 (50.0)	41 (45.6)	47 (52.2)			
5–6	20 (22.2)	17 (18.9)	19 (21.1)	21 (23.3)			

MMD, moyamoya disease; Neu5Ac, N-acetylneuraminic acid; SD, standard deviation; SBP, systolic blood pressure; DBP, diastolic blood pressure; BMI, body mass index; IQR, interquartile range; WBC, white blood cell; LY, lymphocyte; NEUT, neutrophil; MONO, monocyte; PLT, platelet; ALT, alanine transaminase; ALP, alkaline phosphatase; Glu, glucose; Cr, creatinine; TG, triglyceride; TC, total cholesterol; HDL-C, high-density lipoprotein cholesterol; LDL-C, low-density lipoprotein cholesterol; ApoA, apolipoprotein A; ApoB, apolipoprotein B; Hcy, homocysteine; PNR, platelet-to-neutrophil ratio; SII, systemic immune-inflammation index; MHR, monocyte-to-HDL cholesterol ratio; mRS, modified Rankin Scale. * *p* < 0.05 indicates significant difference.

**Table 5 brainsci-13-00913-t005:** Association between baseline Neu5Ac levels and the risk of MMD and its subtypes.

Neu5Ac	No. of Events (%)	Crude Model		Model 1		Model 2	
		OR (95% CI)	*p*-Value	OR (95% CI)	*p*-Value	OR (95% CI)	*p*-Value
MMD overall							
Continuous	360 (80.2)	1.395 (1.141–1.706)	0.001 *	1.380 (1.125–1.694)	0.002 *	1.528 (1.178–1.983)	0.001 *
Neu5Ac level							
Low (2.45–4.52)	171 (76.3)	1.0 (Ref.)		1.0 (Ref.)		1.0 (Ref.)	
High (4.52–16.25)	189 (84.0)	1.627 (1.016–2.606)	0.043 *	1.628 (0.988–2.681)	0.056	1.882 (1.037–3.413)	0.037 *
Ischemic MMD							
Continuous	258 (74.3)	1.299 (1.059–1.593)	0.012 *	1.265 (1.025–1.561)	0.029 *	1.414 (1.077–1.857)	0.013 *
Neu5Ac level							
Low (2.53–4.45)	120 (69.3)	1.0 (Ref.)		1.0 (Ref.)		1.0 (Ref.)	
High (4.45–16.25)	138 (79.3)	0.591 (0.362–0.963)	0.035 *	0.636 (0.374–1.080)	0.094	0.611 (0.316–1.182)	0.144
Hemorrhagic MMD							
Continuous	102 (53.4)	1.666 (1.294–2.144)	<0.001 *	1.758 (1.343–2.302)	<0.001 *	1.806 (1.306–2.498)	<0.001 *
Neu5Ac level							
Low (2.45–4.68)	38 (40.0)	1.0 (Ref.)		1.0 (Ref.)		1.0 (Ref.)	
High (4.68–13.29)	64 (66.6)	3.000 (1.662–5.414)	<0.001 *	3.420 (1.823–6.416)	<0.001 *	3.154 (1.472–6.757)	0.003 *

Neu5Ac, N-acetylneuraminic acid; MMD, moyamoya disease; OR, odds ratio; CI, confidence interval. * *p* < 0.05 indicates significant difference.

**Table 6 brainsci-13-00913-t006:** Categorical NRI and IDI showing the role of Neu5Ac in clinical risk models.

	AUC (95% CI)	Categorical NRI (95% CI)	*p*-Value	IDI (95% CI)	*p*-Value
MMD overall					
Basic model	0.858 (0.817–0.900)	1.0 (Ref.)		1.0 (Ref.)	
Basic model + Neu5Ac binary	0.869 (0.830–0.907)	0.070 (0.001–0.140)	0.047 *	0.010 (−0.002–0.023)	0.120
Basic model + Neu5Ac quartiles	0.863 (0.824–0.903)	0.062 (−0.007–0.131)	0.081	0.020 (0.003–0.037)	0.024 *
Basic model + Neu5Ac continuous	0.873 (0.836–0.911)	0.124 (0.033–0.214)	0.007 *	0.028 (0.008–0.047)	0.005 *
Ischemic MMD					
Basic model	0.887 (0.851–0.924)	1.0 (Ref.)		1.0 (Ref.)	
Basic model + Neu5Ac binary	0.893 (0.858–0.927)	−0.015 (−0.088–0.057)	0.677	0.004 (−0.005–0.014)	0.387
Basic model + Neu5Ac quartiles	0.890 (0.854–0.926)	0.048 (−0.038–0.135)	0.273	0.011 (−0.004–0.027)	0.154
Basic model + Neu5Ac continuous	0.896 (0.862–0.930)	0.067 (−0.013–0.147)	0.101	0.018 (0.001–0.035)	0.042 *
Hemorrhagic MMD					
Basic model	0.849 (0.793–0.905)	1.0 (Ref.)		1.0 (Ref.)	
Basic model + Neu5Ac binary	0.867 (0.816–0.919)	0.152 (0.029–0.274)	0.015 *	0.034 (0.007–0.061)	0.014 *
Basic model + Neu5Ac quartiles	0.858 (0.804–0.911)	0.166 (0.033–0.298)	0.014 *	0.056 (0.022–0.089)	0.001 *
Basic model + Neu5Ac continuous	0.872 (0.821–0.923)	0.199 (0.074–0.324)	0.002 *	0.065 (0.030–0.100)	<0.001 *

Basic model was adjusted for age, sex, SBP, DBP, BMI, PLT count, ALT, ALP, Glu, urea, Cr, TG, TC, ApoB, Hcy, PNR, SII, and MHR. NRI, net reclassification index; IDI, integrated discrimination improvement; Neu5Ac, N-acetylneuraminic acid; MMD, moyamoya disease; AUC, receiver operating characteristic curve; CI, confidence interval; SBP, systolic blood pressure; DBP, diastolic blood pressure; BMI, body mass index; PLT, platelet; ALT, alanine transaminase; ALP, alkaline phosphatase; Glu, glucose; Cr, creatinine; TG, triglyceride; TC, total cholesterol; ApoB, apolipoprotein B; Hcy, homocysteine; PNR, platelet-to-neutrophil ratio; SII, systemic immune-inflammation index; MHR, monocyte-to-HDL cholesterol ratio. * *p* < 0.05 indicates significant difference.

## Data Availability

The data that support the findings of this study are available from the corresponding author upon reasonable request.

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
