# Peer review of "Correlation of Serum N-Acetylneuraminic Acid with the Risk of Moyamoya Disease"

_brainsci, 2023, doi:10.3390/brainsci13060913_

Round 1

Reviewer 1 Report

Regarding the neogenesis of moyamoya disease(MMD), multiple factors such as genetic and environmental factors are involved. Moreover, as the manuscript mentions, the interaction of genetic and environmental factors can change several metabolic pathways during the development of smoky blood vessels, collateral developing pathways to compensate for the reduced cerebral blood supply due to the stenosis/occlusion of the ICA terminal.

The authors seem to be unique to focus on the identification of specific metabolites to elucidate the pathophysiology and to explore the pathogenesis of MMD.

However, I have some concerns about the manuscript.

Focusing on one molecule, N-acetylneuraminic acid (Neu5Ac), the study seems to be very confirmatory and not mechanistic. Unfortunately, it is prone to be a hypothesis-based study.

Figure 2 shows one of the most important results of this study. Indeed the median values of serum Neu5Ac concentrations in MMD and control subjects were 4.62 μmol/L and 4.33 μmol/L (IQR, 1.63), respectively with statistical differences. However, this is a tiny difference. I wonder what biochemical effects of serum Neu5Ac with tiny change induce the tremendous difference of both the normally healthy and MMD group.

 I would have expected the result of a more comprehensive data-based study since we are keen to know the factor contributing to the discovery of the aetiology of MMD.

Author Response

Response to Reviewer 1 Comments

We are grateful for your positive comments, and we have now tried to address each of the cogent comments and suggestions with revision are as follows:

Point 1: Focusing on one molecule, N-acetylneuraminic acid (Neu5Ac), the study seems to be very confirmatory and not mechanistic. Unfortunately, it is prone to be a hypothesis-based study.

Response 1: We highly appreciate the suggestions from the reviewer. At present, the etiology of MMD is unclear, the pathogenesis is complex, the etiology involves many factors, genetics, immunity, environment and many other aspects, the current exploration of Neu5Ac is preliminary, exploring the possible etiology of MMD. As stated in the original Line 405, although we found a new biomarker of MMD and constructed a good model, we lacked the corresponding basic experiments to explore the role of Neu5Ac in the pathogenesis of MMD. On the one hand, due to the limitation of our project funding, on the other hand, although there are many teams trying to construct cellular and animal models of MMD, there are no mature MMD cells or animal models, and the related basic experiments have a bottleneck. For the above reasons, we cannot further explain the relationship between Neu5Ac and the pathogenesis of MMD, but when conditions permit in the future, we will explore it further.

Point 2: Figure 2 shows one of the most important results of this study. Indeed the median values of serum Neu5Ac concentrations in MMD and control subjects were 4.62 μmol/L and 4.33 μmol/L (IQR, 1.63), respectively with statistical differences. However, this is a tiny difference. I wonder what biochemical effects of serum Neu5Ac with tiny change induce the tremendous difference of both the normally healthy and MMD group.

Response 2: Thank you very much for the opinions for this paper! According to previous studies, the concentration of Neu5Ac as a circulating biomarker in cardiovascular disease was 297 ng/ml in the myocardial infarction group (IQR, 220–374) compared to 207 ng/ml in the healthy group (IQR, 114–276, P<0.001) was statistically significant1, with a concentration of 0.94 μM in the heart failure group (IQR, 0.69–1.40) compared to 0.61 μM in the healthy group (IQR, 0.48–0.79, P<0.001) was also statistically significant2. From both studies we observed that differences between disease group and healthy group, although statistically significant, were not numerically significant and similar to our findings. The current study shows that Neu5Ac acts as a circulating biomarker of a disease, and it is not clear whether the increased concentration is a result of the disease or a link in the disease pathogenesis. Due to our current conditions, we cannot conduct basic experiments for further research, so we focus on creating clinical prediction models to predict disease, and if conditions exist in the future, we will study them in cells and animal models.

  1. Li MN, Qian SH, Yao ZY, et al. Correlation of serum N-Acetylneuraminic acid with the risk and prognosis of acute coronary syndrome: a prospective cohort study. BMC Cardiovasc Disord. Sep 10 2020;20(1):404. doi:10.1186/s12872-020-01690-z
  2. C L, M Z, L X, et al. Prognostic Value of Elevated Levels of Plasma N-Acetylneuraminic Acid in Patients With Heart Failure. Circulation Heart failure. 2021;14(11):e008459. doi:10.1161/circheartfailure.121.008459

Reviewer 2 Report

Dear Authors,

Your manuscript is interesting, but some points need to be modified. 

The references in the text should be added before ending the sentence, thus before the "."

The quality of the images in Figure 2 should be enhanced.

Also, the readers would want to know the applicability of your discoveries in medical and clinical life. 

Overall, the manuscript should be entirely checked for the design required by the journal, especially references-wise. 

Author Response

Response to Reviewer 2 Comments

We are grateful for your positive comments, and we have now tried to address each of the cogent comments and suggestions with revision are as follows:

Point 1: The references in the text should be added before ending the sentence, thus before the "."

Response 1: We highly appreciate the suggestions from the reviewer. Then, we add the references in the text before the end of the sentence.

Point 2: The quality of the images in Figure 2 should be enhanced.

Response 2: We highly appreciate the suggestions from the reviewer. Then, we replaced Figure 2 with a clear version and performed new insertions in the manuscript.

Point 3: Also, the readers would want to know the applicability of your discoveries in medical and clinical life.

Response 3: Thank you very much for the opinions for this paper! In the discussion section we added the applicability of our discoveries in medical and clinical life.

Line 334:

“In medical and clinical settings, serum Neu5Ac can be used as a novel biomarker for the diagnosis of MMD, and targeting Neu5Ac may serve as a novel therapeutic strategy for MMD. Furthermore, the multifactorial clinical prediction model including serum Neu5Ac concentrations developed in this study could be used to predict MMD and its subtypes.”

Point 4: Overall, the manuscript should be entirely checked for the design required by the journal, especially references-wise.

Response 4: Thank you very much for the opinions for this paper! We checked the format of the reference again according to the requirements of the journal, and the changes have been reflected in the Response 1.

Reviewer 3 Report

This is a well-designed and well-performed study showing the correlation of serum N-acetylneuraminic acid with the risk of Moyamoya disease. The description of the cohort and the methods used as well as the presentation of the data are well in general.

However, the text needs to be edited by a native English speaker. Just a couple of examples.

Line  183: Levels of Neu5Ac between HCs and MMD and its subtypes. Levels cannot be "between", the difference  is between groups, not levels/

Line 186: Table 2. Baseline characteristics between HCs and MMD subtypes. Again, characteristics cannot be "between".

The whole text should be thoroughly edited.

The text needs to be edited by a native English speaker.

Author Response

Response to Reviewer 3 Comments

We are grateful for your positive comments, and we have now tried to address each of the cogent comments and suggestions with revision are as follows:

Point 1: However, the text needs to be edited by a native English speaker. Just a couple of examples.

Line 183: Levels of Neu5Ac between HCs and MMD and its subtypes. Levels cannot be "between", the difference is between groups, not levels/

Line 186: Table 2. Baseline characteristics between HCs and MMD subtypes. Again, characteristics cannot be "between".

The whole text should be thoroughly edited. Comments on the Quality of English Language. The text needs to be edited by a native English speaker.

Response 1: We highly appreciate the suggestions from the reviewer. The revised manuscript has been polished by a professional academic polishing company. We believe that the English writing of this version has been greatly improved. The language certificate will be attached together.

Round 2

Reviewer 1 Report

I am not convinced by the conclusion that serum Neu5Ac can be considered a new biomarker for diagnosing MMD and targeting Neu5Ac can be seen as a novel therapeutic strategy for MMD. The manuscript appears to rely too heavily on the results without providing sufficient evidence. It only presents the differences in Neu5Ac serum levels between the control group and MMD patients.

Moreover, there is a lack of data regarding the biochemical function of Neu5Ac and the underlying pathogenesis of MMD, such as ICA terminal stenosis. The manuscript seems descriptive rather than conclusive, lacking robust methods and data.

I have the impression that the manuscript tends to confirm existing hypotheses rather than explore new possibilities. If the authors are genuinely interested in understanding the aetiology of MMD, I wonder why they solely focus on the difference in Neu5Ac levels between the control and MMD groups instead of conducting a comprehensive analysis based on multiple data points.

I am curious whether other factors are being considered as potential candidates, such as TG, TC, HDL-C, LDL-C, ApoA, ApoB, Hcy, PNR, SII, and MHR, in addition to Neu5Ac. These factors also demonstrate statistically significant differences between the control and MMD groups
